# Outcomes Associated with a Single Joystick-Operated Ride-on-Toy Navigation Training Incorporated into a Constraint-Induced Movement Therapy Program: A Pilot Feasibility Study

**DOI:** 10.3390/bs13050413

**Published:** 2023-05-15

**Authors:** Sudha Srinivasan, Nidhi Amonkar, Patrick Kumavor, Kristin Morgan, Deborah Bubela

**Affiliations:** 1Physical Therapy Program, Department of Kinesiology, University of Connecticut, Storrs, CT 06268, USA; nidhi.amonkar@uconn.edu (N.A.); deborah.bubela@uconn.edu (D.B.); 2Institute for Collaboration on Health, Intervention, and Policy (InCHIP), University of Connecticut, Storrs, CT 06268, USA; 3The Institute for the Brain and Cognitive Sciences (IBACS), University of Connecticut, Storrs, CT 06268, USA; 4Biomedical Engineering Department, University of Connecticut, Storrs, CT 06268, USA; patrick.d.kumavor@uconn.edu (P.K.); kristin.2.morgan@uconn.edu (K.M.)

**Keywords:** joystick-operated ride-on-toys, children with hemiplegic cerebral palsy, novel technologies for rehabilitation, upper extremity function, wrist-worn accelerometers

## Abstract

Our research aims to evaluate the utility of joystick-operated ride-on-toys (ROTs) as therapeutic adjuncts to improve upper extremity (UE) function in children with hemiplegic cerebral palsy (HCP). This study assessed changes in affected UE use and function following a three-week ROT navigation training incorporated into an existing constraint-induced movement therapy (CIMT) camp in 11 children (3–14 years old) with HCP. We report changes in scores on the standardized Shriners Hospital Upper Extremity Evaluation (SHUEE) from pretest-to-posttest and changes from early-to-late sessions in percent time spent by the affected arm in: (a) “moderate-to-vigorous activity”, “light activity” and “no activity” bouts based on accelerometer data and (b) “independent”, “assisted”, and “no activity” bouts based on video data. We also explored relationships between standardized measures and training-specific measures of affected UE activity. We found small-to-medium improvements in the SHUEE scores. Between 90 and 100% of children also showed medium-to-large improvements in affected UE activity from early-to-late sessions using accelerometers and small improvements via video-based assessments. Exploratory analyses suggested trends for relationships between pretest-posttest and training-specific objective and subjective measures of arm use and function. Our pilot data suggest that single joystick-operated ROTs may serve as motivating, child-friendly tools that can augment conventional therapies such as CIMT to boost treatment dosing, promote affected UE movement practice during real-world navigation tasks, and ultimately improve functional outcomes in children with HCP.

## 1. Introduction

Children with hemiplegic cerebral palsy (HCP) have impaired upper extremity (UE) function on one side of the body with significant hand involvement that leads to considerable limitations in their ability to engage with and learn from their environment [1,2]. Over the last few decades, intensive research on treatment paradigms for improving UE function in children with HCP has led to expanding evidence in favor of task-oriented and intensive approaches [3]. One such evidence-based approach that has evolved out of the adult stroke literature but has also proven to be effective in children with HCP is called constraint-induced movement therapy (CIMT [4,5,6,7,8]. This paradigm involves constraining the child’s unaffected side and encouraging repetitive practice of using the affected UE through structured and intensive UE therapies. Although there is considerable variation within the literature on CIMT in terms of dosing parameters (i.e., duration in weeks of the CIMT program and duration of hours per day of constraint and intensive UE therapy), effective CIMT programs require highly intensive and repetitive active practice using the affected UE during goal-oriented activities [8,9,10,11]. While high dosing is critical to producing meaningful improvements in function through CIMT, clinicians frequently struggle to design activities that children find intrinsically motivating and that will promote sustained adherence with therapy [12,13,14,15]. In fact, therapists and researchers have long recognized that child motivation is related to gains in function and long-term compliance with therapy [16,17]. Children are more likely to practice activities that they find fun and that are aligned with their interests [18]. Therefore, there is a need to diversify conventional therapeutic activities to include novel training ideas and tools that are child-friendly, promote sensorimotor exploration and affected UE function, and encourage UE practice as part of children’s daily play/routines within their naturalistic settings.

Our research team has been exploring the use of modified, commercially available joystick-operated ride-on-toys (ROTs) as therapeutic adjuncts to promote affected UE use and function in children with HCP. Powered ROTs with modified controls (e.g., hand-operated switches instead of leg pedals) have been used previously as early mobility solutions for young children with lower limb impairments, including children with CP and Down syndrome [19,20]. Their use among non-ambulatory children has led to improvements in mobility, social skills, and overall participation [20,21,22,23,24]. However, the use of ROTs to promote UE function has not been explored. We propose that joystick-operated ROTs may serve as engaging adjuncts to conventional care that can be used by clinicians and caregivers to increase treatment dosing and promote children’s functional use of their affected UE for goal-directed navigation within a variety of indoor and outdoor naturalistic settings.

This paper is the third in a series of manuscripts that report data from a pilot study exploring the feasibility of implementation and preliminary efficacy of the ROT training integrated into a three-week CIMT-based camp to promote affected UE use/function among children with HCP. Previously, we reported that the ROT training was feasible to implement within the camp setting and was well-received by children, caregivers, and clinicians. Children expressed the desire to repeat the program and both caregivers and clinicians reported observing improvements in children’s use of their affected UEs as well as motor function following the training [25]. In the second paper, we report improvements in video-based measures of arm control and navigational accuracy following the ROT training provided within the CIMT camp ([26] under review). In the present manuscript, we report the combined effects of the ROT training and CIMT activities on objective and subjective measures of affected UE function. Specifically, we will report on changes in movement quantity (measured using wrist-worn accelerometers) and quality (assessed using standardized tests and training-specific measures) following the training. We will also explore relationships between affected UE use/control during ROT operation within training sessions and hand-use during everyday functional activities outside the training context.

We hypothesize that children will show improvements in affected UE use and motor function as assessed using qualitative and quantitative measures within the ROT training context as well as outside the training sessions. Moreover, we hypothesize that training-specific measures of motor function will show trends for associations with children’s motor performance outside the training context on a standardized test.

## 2. Materials and Methods

### 2.1. Participants

Eleven children with HCP (6M, 5F; 8 children with right-sided involvement and 3 children with left-sided involvement) between 3 and 14 years (mean (SD): 6.54 (2.76); 7 Caucasian, 1 Hispanic, 3 of mixed ethnicity), with a moderate level of impairment (mean (SD): 2.64 (0.67), see Table 1 for scores on the manual ability classification system (MACS) [27] participated in the single group pre–post study. The study was conducted within a three-week CIMT-based summer camp. The study was approved by the Institutional Review Board (IRB) at the University of Connecticut, Storrs.

The single joystick-operated ride-on-toy navigation training was incorporated into the Lefty and Righty Camp of Connecticut (LARC), an annually held summer camp for children with HCP, based on principles of the CIMT. The camp activities were designed to provide children with playful movement experiences to improve gross and fine motor function of the affected UE. The ride-on-toy training was offered as one of the daily activities at camp for each child (see details of the camp and the ROT training within the section on procedures). Parental permission and child written/oral assent were obtained prior to any testing or training procedures.

### 2.2. Outcome Measures and Materials

#### 2.2.1. Pretest–Posttest Measures of Motor Function

The Shriners Hospital Upper Extremity Evaluation (SHUEE) is a standardized, valid, and reliable test to assess movement quality in 3–18-year-old children (inter-rater reliability: 0.89–0.90 (ICC), intra-rater reliability: 0.98–0.99 (ICC)) [28]. The test assesses affected UE use spontaneously and on tester demand during 16 bimanual tasks [29]. The test has 3 parts: spontaneous functional analysis (SFA), dynamic positional analysis (DPA), and grasp-release analysis. At present, there are no data available on standard error of measurement or minimal clinically important difference (MCID) for the SHUEE. However, the SHUEE has been used to assess the efficacy of surgical interventions with children with HCP in multiple studies [30,31,32,33]. For this study, we analyzed changes in the total SFA scores (i.e., child’s ability to spontaneously use the affected UE during bimanual tasks) and total DPA scores (i.e., segmental alignment of the affected UE at the elbow, forearm, wrist, fingers, and thumb while performing tasks on demand) from pretest (prior to camp) to posttest (following the camp). A single coder coded all the data after establishing intra-rater reliability and inter-rater reliability (with the first and second authors) of over 90% using 20% of the videos.

#### 2.2.2. Training-Specific Measures Assessed during Early and Late Sessions

##### Objective Accelerometry-Based Assessment of Affected UE Activity during ROT Navigation

Children wore the wGT3X-BT accelerometers (ActiGraph, Pensacola, FL, USA) on the wrist of their affected arm during the entire duration of the ROT training sessions in the first and last weeks of the training program. The wGT3X-BT accelerometer is a small (4.6 × 3.3 × 1.5 cm), lightweight (19 g), 3-axis accelerometer that collects raw acceleration data in all 3 directions with a dynamic range of ±8 g (gravitational units). The accelerometers collected data at a sampling frequency of 30 Hz. Trainers maintained activity diaries of the exact times of the training sessions every day for each child to corroborate the data obtained from the accelerometers. For their data to be included within the analysis, children were required to wear the accelerometer during the ROT training sessions on at least 3 training sessions at each time point (early and late training weeks). Since children were seated in the ROT during the training sessions, data collected through the activity monitor is solely representative of affected UE activity during the training sessions.

At the end of the first and last weeks, data stored in the accelerometers for each child was downloaded using the ActiLife software (ActiGraph, Pensacola, FL, USA). The raw data from the accelerometers were processed using ActiGraph’s proprietary algorithms to obtain activity counts (1 count = 0.001664 g, i.e., 0.0163072 m/s^2^). Activity counts across 3 axes were summed to calculate vector magnitude (VM) counts as follows:
VM = √(a_x^2^ + a_y^2^ + a_z^2^),
where a_x, a_y, and a_z are the accelerations in the x-, y-, and z-directions, respectively. We assessed changes in average VM counts (averaged across all training sessions during a week) across early and late training weeks. Moreover, the in-built, Freedson children algorithm was used to classify average activity counts calculated over 60 s epochs during ROT navigation sessions across the entire week into time spent (in minutes) by the affected UE in activities of varying intensity (sedentary: 0–149 counts, light activity: 150–499 counts, and moderate-to-vigorous activity: >500 counts) [30]. The minimal clinically important difference (MCID) for the arm accelerometry is a change of 575–752 counts [31]. Please note that MCID values for arm accelerometry are based on data from adult persons with chronic hemiparesis since no similar data are available from children with HCP. We report changes in the average percent time spent by the affected UE in sedentary, light, and moderate-to-vigorous activity during ROT navigation sessions across early and late training weeks.

##### Video-Based Assessment of Affected UE Activity during ROT Navigation

Video data of early and late training sessions were coded using Datavyu© behavioral coding software that allows millisecond-to-millisecond coding of behaviors. A single coder coded all data after establishing intra-rater and inter-rater reliability (with a second coder) of over 90% using a subset of videos (20%) from the study. We coded affected UE activity during ROT navigation based on video data from 2 early and 2 late training sessions. Specifically, each ROT session was broken down into time blocks of “independent”, “assisted”, and “no activity” bouts. “Independent” activity bouts were defined as periods when the child independently maneuvered the joystick of the ROT using their affected UE without any assistance from an adult/external aid. “Assisted” activity bouts included instances where the child required assistance for controlling the joystick with their affected UE. The assistance could be in the form of an external aid (such as a mitt) to help the child grasp the joystick or the adult trainer providing partial or total assistance to help the child push/maneuver the joystick. “No activity” bouts included periods when the child was stationary, and the affected UE was not used to maneuver the joystick of the ROT. We report on the average percent duration of time of independent, assisted, and no activity bouts in the affected UE during early and late training sessions.

### 2.3. Procedures

#### 2.3.1. Camp Structure and Activities

The three-week intensive, 6 h/day (9 am to 3 pm) summer camp provided group-based CIMT for children with HCP. During the daily six hours at camp, all children wore removable thermoplastic casts on their unaffected UE. Children were encouraged to use their affected side throughout the day during goal-directed gross and fine motor activities/games, as well as functional self-care tasks such as eating and toileting. Each child worked one-on-one with a camp staff who was a trained paraprofessional under the supervision of licensed physical and occupational therapists.

#### 2.3.2. Ride-on-Toy Training Program

The ROT training was incorporated into the camp routine and was offered as one of the daily activities at camp. Each ROT session lasted for around 20–30 min/day. Please note that children received an overall 90 h of CIMT (6 h/day, 5 days/week, 3 weeks) at camp, of which 8 h involved ROT training. Our research team modified a commercially available, dual joystick-operated ROT, the Wild Thing^TM^, to allow operation in a single joystick mode and provided additional postural support (using PVC pipes for reinforcement of the external frame of the toy; see Figure 1). As part of the ROT training program, children engaged in: (a) incrementally challenging navigation games across different environmental layouts and (b) gross and fine motor UE tasks at intermediate stations along the navigational path. To drive the ROT, children were required to use their affected UE to push/pull and maneuver the joystick in the desired direction of motion. Early sessions focused on teaching the child basic joystick controls for moving forward–backward and making turns. Thereafter, the training was progressed to challenge children to stay on paths of different shapes and sizes (arc, roundabout, slalom, etc.) and avoid obstacles during navigation. Children also completed UE gross and fine tasks at intermediate stations during navigation; the tasks involved multidirectional reaching, catching and throwing objects, different grasps, release, and in-hand manipulation of playful props such as balls and bean bags. The training was based on principles of motor learning and promoted discovery learning, variable practice, active problem-solving, and free play/exploration.

We focused on promoting functional UE movement patterns during the ROT navigation program. Grasp and operation of the joystick required wrist extensor, finger flexor, and hand intrinsic muscles while the forearm was maintained in pronation. In addition, children used proximal muscles at the elbow and shoulder to control push–pull movements of the joystick in all 4 directions (forward, backward, right turn, and left turn). In our experience, children with poor UE control tend to also use proximal scapular and trunk muscles to move the joysticks. As discussed above, we also incorporated a variety of functional UE tasks within the training program. Children performed these tasks while seated in the ROT at intermediate checkpoints/stations during navigation. These UE tasks involved gross motor activities such as reaching in different directions, overhead throwing, pulling, pushing, lifting, and tossing games as well as fine motor activities such as opening and closing, precision grips, picking, sticking, and releasing objects. While singular joint movements that are typically limited in HCP (forearm supination, wrist extension, and finger extension) were not addressed in isolation, these movements were encouraged as part of multi-joint movement patterns as children engaged in functional UE challenges/games throughout the ROT program.

### 2.4. Statistical Analyses

Data were checked for assumptions of parametric statistics. Since data satisfied the assumptions of parametric statistics, we used dependent *t*-tests to assess training-related changes in the standardized SHUEE from the pretest to posttest. We conducted Pillai’s trace multivariate analyses of variance (MANOVAs) to evaluate changes in the training-specific measures: (a) wrist-worn accelerometry-based outcomes and (b) video-based estimates of affected UE activity. The MANOVA for accelerometry-based measures included time (early and late sessions) and affected UE activity (percent time spent in sedentary, light, and moderate-to-vigorous activity) as within-subjects factors. The MANOVA for video-based measures included time (early and late sessions) and affected UE activity (percent time engaged in independent, assisted, and no activity bouts) as within-subjects factors. If the analyses found a significant main effect and an interaction effect involving the same factors, post hoc *t*-tests were conducted to evaluate only the significant interactions. We used dependent *t*-tests to assess training-related changes in average VM counts/minute. Statistical significance was set at a *p*-value of <0.05. Effect sizes were calculated using Hedge’s standardized mean difference (SMD) [32]. We report on SMD estimates and 95% confidence intervals (CI) surrounding the SMD values. We classified SMD values according to Cohen’s conventions of small (0.2–0.49), medium (0.5–0.79), or large (0.8 and above) effects [33].

We also conducted exploratory analyses to evaluate trends for associations between pretest–posttest measures and assessments administered during training sessions (accelerometry and video-based coding of affected UE use). We have used scatter plots to visually represent patterns of associations between measured variables using both pooled data (pooled across early and late sessions or pretest and posttest) and difference data (late–early session or posttest–pretest values). Given the small sample size in this pilot exploratory study, we will not conduct formal tests of significance for these plotted correlations between variables; instead, we interpret the visual data as being suggestive of preliminary trends for relationships between variables that we will confirm in our future studies using larger sample sizes and more robust study designs.

## 3. Results

### 3.1. Pretest–Posttest Measures of Affected UE Motor Function

From pretest to posttest, children showed significant medium-sized increases in the SFA scores on the standardized SHUEE (see Figure 2A,B; *t*(10) = 4.114, *p* = 0.002, SMD (95% CI) = 0.5 (−0.21 to 1.21)), with 10 out of 11 children following the group trend. All 11 children also showed small-to-medium increases in DPA scores (see Figure 2A,C; *t*(10) = 5.977, *p* ≤ 0.001, SMD (95% CI) = 0.301 (−0.37 to 0.98)), specifically, in the positioning of the elbow (*t*(10) = 2.324, *p* = 0.042, SMD (95% CI) = 0.445 (−0.25 to 1.41)), forearm (*t*(10) = 3.184, *p* = 0.010, SMD (95% CI) = 0.302 (−0.37 to 0.98)) and wrist in the sagittal plane (*t*(10) = −2.390, *p* = 0.038, SMD (95% CI) = 0.152 (−0.51 to 0.81)). Overall, children showed a mean improvement of 9.29% on the SFA and 7.45% in total DPA scores.

### 3.2. Objective Accelerometry-Based Measures of Affected UE Activity during Training Sessions

The overall adherence rate with accelerometer wear was 100% and all children wore the monitor on their affected UE for a minimum of three ROT sessions during the week with no complaints. The MANOVA for the intensity of affected arm activity indicated a significant main effect of time (*F* (2, 9) = 105.52, *p* < 0.001, $n_{p}^{2}$ = 0.95) and an interaction effect of UE activity × time (*F* (2, 9) = 13.48, *p* = 0.002, $n_{p}^{2}$ = 0.750). Post hoc analyses of the significant interaction effect suggested that from early-to-late sessions, the percent time spent by the affected UE in light activity decreased by a large effect size (SMD (95% CI): −0.88 (−1.70 to −0.07)), with a concurrent medium-sized increase in time spent in moderate-to-vigorous activity (see Figure 3A, SMD (95% CI) = 0.72 (−0.04 to 1.49)).

Specifically, 10 out of 11 children decreased the time spent in light activity and all 11 children increased time spent in moderate-to-vigorous arm activity from early-to-late sessions (see Figure 3B,C). Children also showed a statistically significantly large increase in average VM counts/minute from early-to-late sessions (mean (SD): early: 1793.67 (790.7), late: 2490.06 (870.8); SMD (95% CI) = 0.81 (0.018 to 1.6)), with 10 out of 11 children, following these group trends. Overall, children showed a mean increase in VM counts of 696.39 counts/minute following training.

### 3.3. Observational Video-Based Assessment of Affected UE Activity during Training Sessions

The MANOVA indicated a significant main effect of affected UE activity (F (2, 9) = 17.13, *p* < 0.001, $n_{p}^{2}$ = 0.981) and an interaction effect of affected UE activity × time (F (2, 9) = 2.79, *p* = 0.001, $n_{p}^{2}$ = 0.770). Post hoc testing of the significant interaction suggested that from early to late sessions, there was a significant small increase in the percent duration of “independent” UE activity bouts (SMD (95% CI) = 0.20 (0.19 to 1.95)) and a concurrent decrease in percent time spent by the affected UE in “assisted” activity bouts (SMD (95% CI) = −0.15 (−1.49 to 0.04)) and “no activity” bouts (SMD (95% CI) = −0.16 (−1.61 to −0.02)) (See Figure 4A–C).

### 3.4. Exploratory Analyses of Associations between Pretest–Posttest and Training-Specific Measures of Affected UE Activity

We used scatter plots to explore relationships between standardized and training-specific variables using pooled data (pooled across pretest and posttest or early and late sessions, see Figure 5A) and difference data (i.e., differences between posttest and pretest or late and early session performance; see Figure 5B). For these analyses, please note that we had one child who showed a large improvement with training; this child required complete assistance on their affected side to begin with, but their active use of the affected UE increased over the course of the training. This child’s data are visually clearly separated from the rest of the group in some of the graphs (see Figure 5A,B). We therefore conducted exploratory analyses both with and without data from this child. We only report on data trends that showed similar patterns (in terms of direction and magnitude of associations) both with and without this child’s data. In other words, we further discuss only preliminary associations between variables that were consistently observed across a majority of the children in the study. For the pooled data, we found a trend for a negative relationship between SHUEE SFA and DPA scores and time spent in assisted navigation, suggesting that children who were less likely to require assistance during ROT navigation had higher SHUEE scores. Moreover, our pilot data suggest preliminary associations between accelerometry-based quantitative measures and video-based subjective measures of affected UE use for navigation (see Figure 5A). Specifically, time spent in moderate-to-vigorous arm activity showed a trend for being positively related with “independent” activity bouts and negatively related with “assisted” navigation (see Figure 5A). On the other hand, time classified as sedentary was related positively with “assisted” mobility bouts. Overall, these exploratory trends suggest that children who were able to drive the ROT independently using their affected UE indicated by video data tended to demonstrate higher levels of moderate-to-vigorous activity with their affected UE and lower levels of sedentary time as measured by wrist-worn accelerometers (see Figure 5A).

Exploratory scatter plots visualizing relationships between variables for difference data (posttest-pretest or late-early sessions) suggested a trend for improvements in SHUEE scores (from pretest to posttest) to be associated with an increase in “independent” activity and a decrease in “no activity” bouts (from early to late sessions) based on video data, as well as a concurrent decrease in percent time spent in light activity as measured by accelerometers from early to late sessions (see Figure 5B).

## 4. Discussion

### 4.1. Summary of Results

Our pilot study suggested that a short three-week ROT navigation training combined with a CIMT program contributed to improvements in affected UE use and motor function during the ROT training sessions with functional carryover outside the training context. Moreover, children showed improvements in both subjective and objective measures that assessed UE movement quality and quantity. Based on our exploratory analyses, we also found trends for associations between measures of UE function assessed during training sessions and a structured standardized test of motor function administered at the pretest and posttest. Moreover, quantitative and qualitative training-specific measures also seemed to be associated with each other, suggesting a corroboration between video-based data and wrist-worn accelerometry to assess affected UE activity in children with HCP. Overall, in conjunction with our previous work in this area [25,26], our findings suggest that joystick-operated ROTs incorporated into a CIMT protocol can serve as effective and child-friendly training tools to promote use of the affected UE among children with HCP. Next, we briefly discuss our findings in the context of the existing literature and the implications of this work.

### 4.2. Training-Related Changes in Affected UE Activity and Motor Function with ROT Navigation Training

A recent review that assessed the minimum threshold dosing necessary to produce meaningful functional gains in affected UE function among children with HCP indicated that more than 30–40 h of goal-directed functional training is required to produce meaningful improvements [34]. Moreover, the authors also acknowledged that beyond dosing, enjoyment and motivation are key factors that influence outcomes, and that the incorporation of home practice as a supplement to face-to-face therapy is a cost-effective solution to enhance therapeutic success [34,35,36]. Our goal with this line of research is to explore the utility and efficacy of joystick-operated ride-on-toys as easy-to-use, cost-effective, and intrinsically motivating tools that can be used by clinicians and families to augment dosing/practice of goal-oriented activities and lead to gains in UE function through experience-dependent neuroplastic processes. The novelty of our approach lies in the choice of an unconventional yet age-appropriate and motivating activity to diversify existing activity choices used in CIMT paradigms. It was encouraging to see that ROT combined with conventional therapy led to not only improved navigational skills within the training context, but also the carryover of motor improvements to a standardized functional test outside the training context.

Improvements in movement quantity and quality may be attributed to the nature of the ROT navigation program. Real-world navigation requires a coupling of perceptual, action-based, and cognitive systems, as the child plans their route in space, adapts to changing environmental and task constraints (e.g., different surfaces, obstacles along the path, changes in elevation in the form of slopes), and skillfully maneuvers the joystick in an adaptive manner to move through their physical environment. We observed that children initially required more assistance to control the joystick; however, over the course of the training, children became more independent and engaged in more purposeful and controlled movements of their affected UE to push the joystick. Even children who were more severely involved with limited voluntary control on the affected UE developed new synergies to use proximal shoulder and trunk muscles and body mechanics to achieve success and independence in navigation. Our findings align with a study by Weightman and colleagues that assessed joystick-control abilities in children with HCP and typically developing children; the authors found that children with HCP tend to rely more on their proximal trunk and shoulder muscles compared to their neurotypical peers while operating the joystick [37]. Children in our study persisted with the training activity in order to achieve functional success in self-driven mobility within the environment. The training effects may also be attributed to the incorporation of motor learning principles into our protocol, including variable and repetitive practice, the provision of multimodal feedback and reinforcement, the use of progressively challenging activities that provide a “just-right” challenge, and fostering free play and exploration [38,39,40,41]. Other studies have also identified that interventions based on motor learning principles are effective in producing improvements in neuromotor function among children with disabilities [42,43,44].

Our findings are also in line with studies that used novel activity ideas and technologies to encourage goal-oriented UE practice among children with HCP [45,46,47,48,49,50,51,52,53,54,55,56]. For instance, Spencer and colleagues used magic-themed activities to incentivize the use of the affected UE in children with HCP to perform magic tricks [50,52,53,57]. They found that a short-term magic-themed camp led to improvements in UE function among children [52]; moreover, children and parents perceived the activities to be fun and motivating while promoting learning and children’s willingness to persist with relatively challenging UE activities [52,53]. Other studies have used more high-tech and immersive tools such as virtual reality and robotics to promote the repetitive practice of progressively challenging activities. For instance, Fluet et al. (2010) found small-to-large improvements in the active movements of shoulder abduction, shoulder flexion, and forearm supination, as well as in several measures of reaching kinematics following a short three-week robot-assisted therapy program for children with HCP [49]. Similarly, Acar and colleagues found greater improvements in hand function and movement speed on the standardized Jebsen–Taylor hand function test in a group of children with CP that received six weeks of biweekly Nintendo Wii virtual reality training sessions in addition to conventional therapy compared to a group that only received conventional physical therapy [46]. Overall, our study adds to the body of literature exploring innovative, child-preferred, and diverse types of activities and tools that can be used to incentivize affected UE use and goal-directed practice both as part of conventional therapy as well as outside conventional therapy contexts. By developing activities that align with children’s interests and goals, we can maximize their engagement in rehabilitation and overall self-confidence.

Among the types of assessments we used, we found the largest effect sizes for improvements in accelerometer-based estimates of affected UE activity compared to changes in standardized and video-based tests. Improvements in VM counts/minute were greater than the minimal clinically important difference values for arm accelerometry reported in the literature [31], suggesting that the improvements were meaningful and may be reflective of functional changes. Accelerometers have been validated as an outcome measure for assessing the efficacy of behavioral interventions both in adults and children with hemiplegia [58,59,60,61]. Accelerometers offer the advantage of being lightweight, non-intrusive, and compact. Moreover, unlike standardized or video-based assessments that provide information on the child’s motor performance based on a single snapshot of time often within a structured setting, accelerometers allow the assessment of children’s habitual UE activity over long durations across a variety of settings and tasks, therefore reflecting the real-world use of their affected UE. In our study, accelerometer-based measures seemed to be more sensitive to capturing changes in affected UE use and intensity of activity from early to late training sessions. We recommend that clinicians use accelerometers as a sensitive outcome measure to assess the effectiveness of behavioral interventions in promoting affected UE use for activities of daily living as part of the child’s daily routines.

### 4.3. Exploratory Associations between Pretest–Posttest and Training-Specific Measures of Affected UE Activity

Our exploratory analyses suggested some interesting trends for relationships between training-specific variables and a standardized test of motor performance. Although promising, these results must be interpreted with caution given our small sample size and the pilot nature of our study. These associations need to be replicated using larger sample sizes and more rigorous study designs. We found trends for associations between spontaneous use of the affected UE during bimanual tasks on the SHUEE and independent and active use of the affected UE during training sessions. These data provide pilot evidence that improvements within the ROT training context may be related to positive improvements on standardized functional tests conducted outside the training context. Our findings are in line with other studies that evaluated associations between affected UE use as assessed using accelerometry and standardized tests of motor performance [4,59,62]. For instance, Sokal and colleagues reported medium-sized correlations between the intensity of use of the affected arm and a standardized test of motor capacity in children with HCP [62].

Moreover, accelerometer-derived measures of affected UE activity were also related to video-based measures of affected UE activity. Our findings are aligned with other studies that also explored associations between accelerometer-based outcomes and performance on standardized tests and video-based assessments in children and adults with neurological impairments [63,64,65]. For instance, Poitras and colleagues found good agreement between video-based and accelerometry-based measurements of arm movements during 20 min of free play in a seated position in adults with CP [65]. Similarly, Uswatte et al. found high correlations between threshold-filtered ActiGraph recordings of UE activity and observer ratings of video data in adult patients with unilateral weakness [63]. Our preliminary data add to the previous literature in support of the use of accelerometers as useful outcome measures for assessing UE activity. Video-based measures may be limited to short durations of capture time, limited capture volume, and tedious and time-consuming efforts to code collected video data; in contrast, accelerometer-based estimates offer longer data collection times across a variety of settings without a significant setup, relatively simple postprocessing, and provide accurate and sensitive objective measures of affected UE activity [4,58,66,67]. Overall, we recommend that accelerometers can be used to supplement clinician observations within therapeutic settings and may serve as a sensitive measure to assess the effects of short-term training programs such as a ROT navigation program aimed to improve UE function.

### 4.4. Limitations & Future Directions

Our study is limited due to the lack of a control group, a small convenience sample, including children with HCP without any history of recent surgeries, a wide age range of ability levels within the participants, lack of sensory testing measures, and the lack of follow-up testing. In this study, we only reported on motor outcomes. However, children with HCP also have sensory impairments that may contribute to their clinical presentation. In our future studies, we will also include measures of sensory function within our test battery. We incorporated our training within a CIMT paradigm; an inherent limitation of CIMT is that it does not allow for mirror movements involving the unaffected UE. The data collected from the accelerometers was analyzed using algorithms that were validated for neurotypical children as there were no validated algorithms available for children with HCP. Moreover, we did not collect accelerometer data outside the training context and at follow-up after completion of the ROT camp. Despite reporting improvements in UE activity during and following ROT navigation when combined with the CIMT program, we were not able to isolate the effects of the ROT training from other camp activities. Our findings also cannot be directly generalized to children with HCP who have undergone surgical procedures such as Botox injections or tendon transfers. Finally, we conducted exploratory analyses that examined trends for relationships between standardized and training-specific variables; these patterns need to be verified using larger sample sizes. In our future studies, we will address some of these limitations by assessing the isolated short-term and long-term effects of a community-based ROT training program using controlled designs and larger homogenous samples.

## 5. Conclusions

Our pilot study assessed the effects of a three-week joystick-operated ROT navigation training program incorporated as part of a CIMT summer camp on spontaneous use, motor function, and activity of the affected UE in children with HCP using a standardized test, wrist-worn accelerometry, and video-based measures. We found improvements on the standardized test of motor function from pretest to posttest as well as increased activity of the affected UE during ROT navigation from early to late training sessions as indicated by accelerometers and video-based measures of navigation. We also found trends for associations between training-specific and standardized measures of UE activity as well as between quantitative and qualitative measures of affected UE use. Joystick-operated ROTs seem to be effective and child-friendly tools that can be easily incorporated into children’s play and conventional therapy by caregivers and clinicians to boost treatment dosing and encourage children with HCP to use their affected UE for purposeful navigation through real-world environments. Our findings have implications for the use of ROTs as therapy adjuncts in UE rehabilitation for children with motor disabilities.

## Figures and Tables

**Figure 1 behavsci-13-00413-f001:**
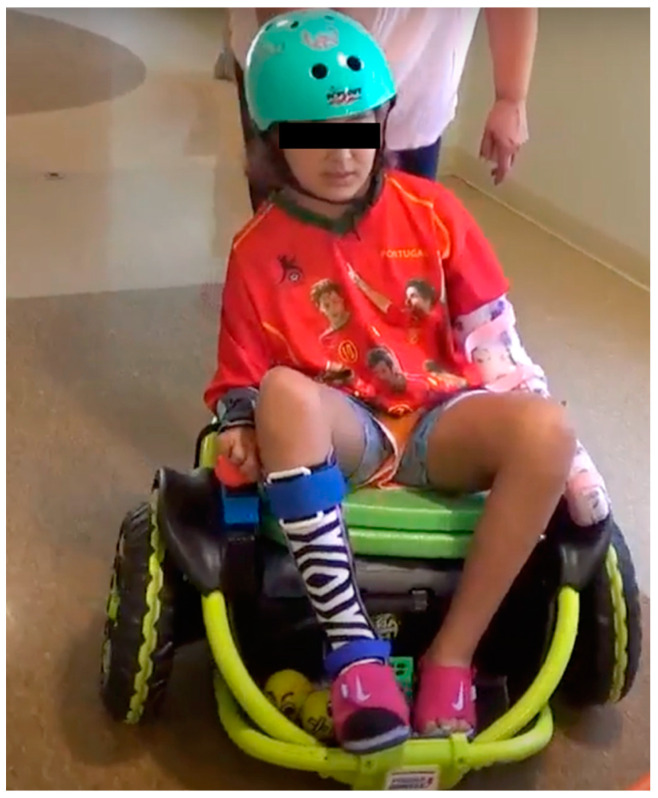
The Wild Thing^TM^ ride-on-toy used for the study.

**Figure 2 behavsci-13-00413-f002:**
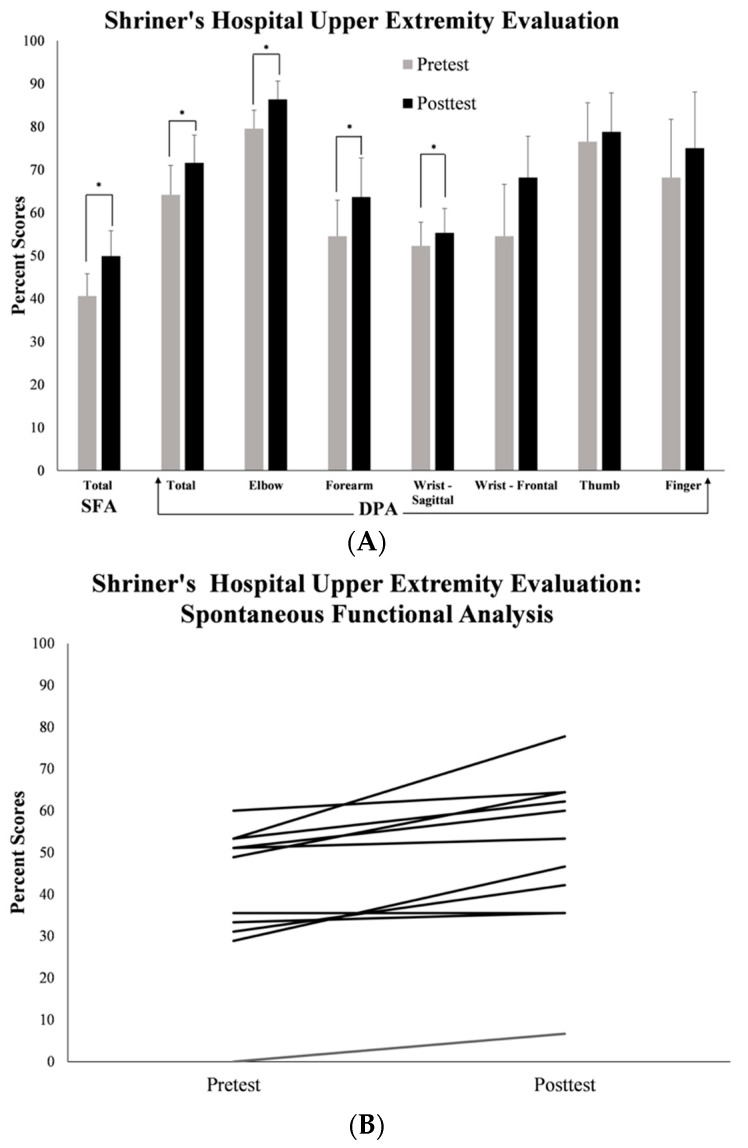
(**A**) Group data on training-related changes on the SHUEE assessed before and after the three-week ROT navigation + CIMT training. (**B**) Individual data on training-related changes in the SFA scores of the SHUEE assessed before and after the three-week ROT navigation + CIMT training. (**C**) Individual data on training-related changes in total DPA scores of the SHUEE assessed before and after the three-week ROT navigation + CIMT training. Please note that * signifies statistically significant differences in measured outcomes at *p* ≤ 0.05.

**Figure 3 behavsci-13-00413-f003:**
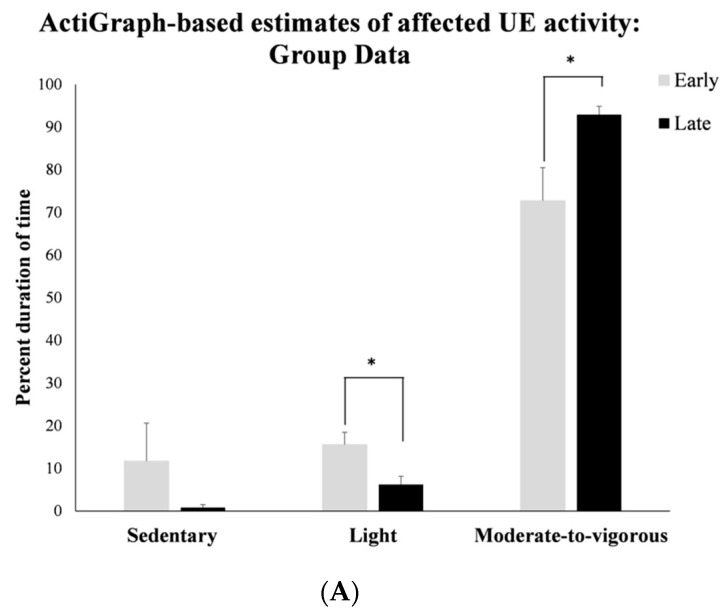
(**A**) Group data on training-related changes in affected UE activity during ROT navigation training measured using wrist-worn accelerometers. (**B**) Individual data on training-related changes in “moderate-to-vigorous” activity in the affected UE during ROT navigation measured using wrist-worn accelerometers (note: the one child who showed a large increase in activity from early-to-late session had to be completely assisted by the trainer in the early session but was able to initiate independent UE activity to push the joystick by the late training weeks). (**C**) Individual data on training-related changes in “light” activity in the affected UE during ROT navigation measured using wrist-worn accelerometers. Please note that * signifies statistically significant differences in measured outcomes at *p* ≤ 0.05.

**Figure 4 behavsci-13-00413-f004:**
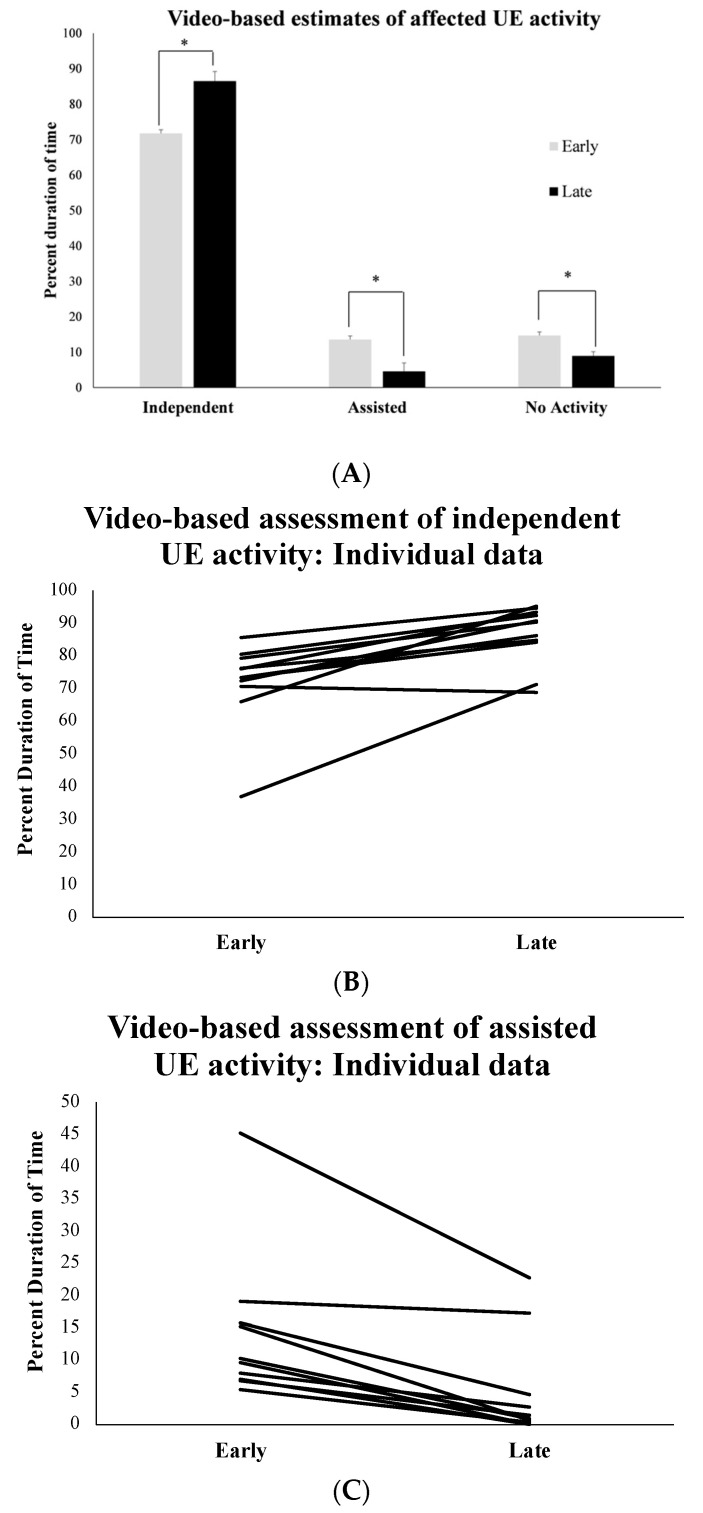
(**A**) Group data on training-related changes in affected UE activity during ROT navigation based on expert ratings of video data. (**B**) Individual data on training-related changes in “independent” use of the affected UE during ROT navigation as measured by video-based observational coding. (**C**) Individual data on training-related changes in “assisted” use of the affected UE activity during ROT navigation as measured by video-based observational coding. Please note that * signifies statistically significant differences in measured outcomes at *p* ≤ 0.05.

**Figure 5 behavsci-13-00413-f005:**
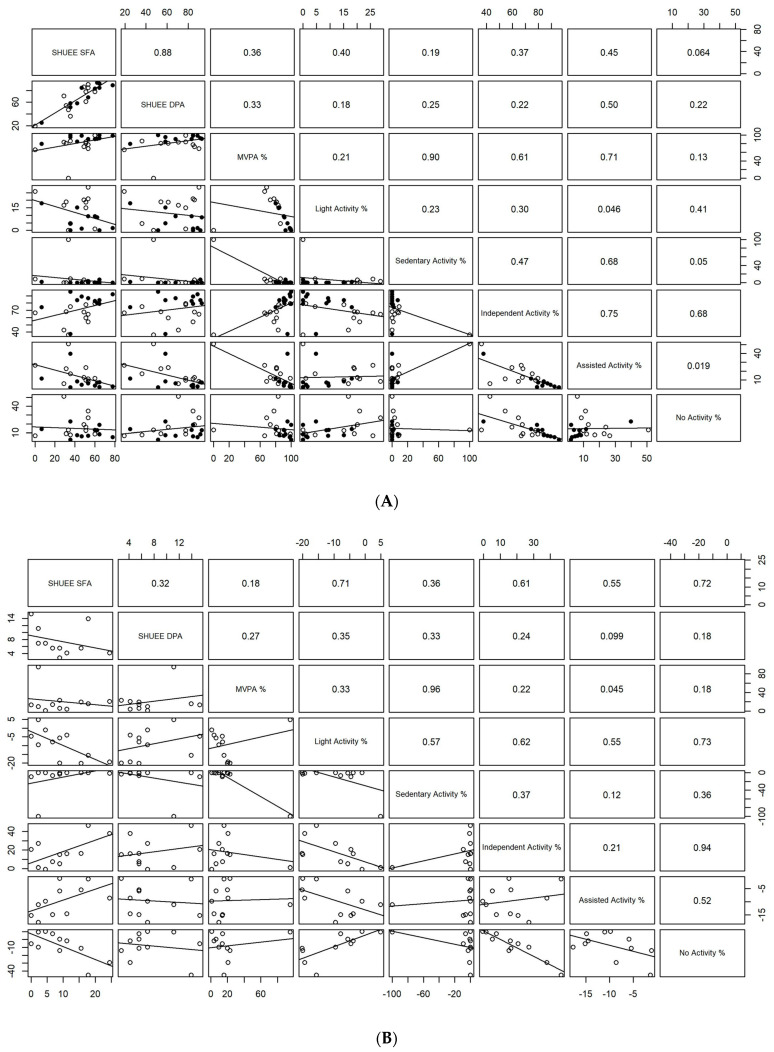
(**A**) Scatter plots of pooled data from standardized and training-specific measures collected in the study. Closed circles represent data from pretest/early sessions and open circles represent data from posttest/late sessions for each individual participant. The individual plots include regression lines fitted to the data. The numerical values are correlation coefficients exploring relationships between measured variables. Note that formal statistical tests of significance for the correlation coefficients have not been conducted due to the small sample size and pilot nature of the study. (**B**) Scatter plots of training-related improvements in standardized and training-specific measures collected in the study. Each open circle represents data from one participant. The individual plots include regression lines fitted to the data. The numerical values are correlation coefficients exploring relationships between measured variables. Note that formal statistical tests of significance for the correlation coefficients have not been conducted due to the small sample size and pilot nature of the study. Note that in both plots, one of the children in the study showed a significantly larger improvement from pretest to posttest compared to the rest of the children in the study. This child required complete assistance from an adult during testing and training activities and showed high levels of sedentary UE activity; however, with training, the child increased the frequency of affected UE activity and independent navigation. Please note that all preliminary trends discussed further within the manuscript hold true even when these analyses were repeated without this child’s data.

**Table 1 behavsci-13-00413-t001:** Demographic details of study participants.

Child Number	Age at Visit	Gender	Race/Ethnicity	Side of Involvement	MACS Levels
1	3 years 5 months	F	White, Non-Hispanic	L	3
2	13 years 10 months	F	Asian	R	3
3	8 years 11 months	M	White, Non-Hispanic	R	3
4	4 years 6 months	F	White, Non-Hispanic	L	2
5	5 years 28 days	M	Multiracial-Korean, Puerto Rican, Irish, and Polish	R	2
6	8 years 3 months	F	White, Hispanic	R	2
7	6 years 11 months	F	White, Non-Hispanic	R	2
8	8 years 7 months	M	White, Hispanic	R	3
9	4 years 2 months	M	White, Non-Hispanic	L	4
10	6 years 11 months	M	Multiracial-White, Asian	R	2
11	7 years 5 months	M	White, Non-Hispanic	R	3

MACS: Manual ability classification system.

## Data Availability

Data from this project will be made available on request to the corresponding author.

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
