# Peer review of "Outcomes Associated with a Single Joystick-Operated Ride-on-Toy Navigation Training Incorporated into a Constraint-Induced Movement Therapy Program: A Pilot Feasibility Study"

_behavsci, 2023, doi:10.3390/bs13050413_

Round 1

Reviewer 1 Report

Review

Can a 3-week single joystick-operated ride-on-toy training program lead to improved spontaneous use and motor function in 3 the affected upper extremity in children with Hemiplegic Cerebral Palsy? Results from a pilot feasibility study

Dear authors

This is a nice example of innovative approach within existing conditions. Thank you for that.

I have some questions, helping to improve the paper.

Because this ROT program is within a 3 weeks CIMT program, the title is not correct.

It should be about the added value of ROT above CIMT.

I assume you do not have results form children without the ROT  program. Because in this program one cannot specify the results  of ROT alone.

This impacts the introduction and especially the discussion and conclusions.

11 children joined. They all performed 90 h of CIMT, of which 8 h are ROT.

One need to discuss intense the added value based on dose (90 versus 8 h).

Please inform us what is known about the psychometric values of the outcome measures ( especially SEM and clinical important change) in the method section and use these values in the results presentation and discussion/ conclusion.

This is a pilot feasibility and to perform this type of analysis and correlation should be avoided in my opinion. I would like to have the opinion of a statistician.

I prefer the description of the individual scores to indicate improvement or not, without the complexity of these type of statistics, with a low value based on the sample size

The paper is written very well.

I hope the authors will look into these topics to improve the paper

Best

Author Response

Dear Editor and Reviewers,

We sincerely thank the reviewers for taking the time to review our manuscript and for providing excellent and constructive feedback. We have addressed all concerns raised by both reviewers. We believe that the manuscript is truly strengthened following these revisions. Below, we provided a point-by-point response to reviewer comments with reference to manuscript edits. All edits in the manuscript have been made using track changes.

We hope we have addressed all the concerns of the reviewers and we look forward to the next round of review.

Thank you very much for your time and consideration.

Sincerely,

Authors

Reviewer comments

Reviewer 1:

Comment 1: Because this ROT program is within a 3 weeks CIMT program, the title is not correct. It should be about the added value of ROT above CIMT.

Response:  We appreciate the reviewer’s comment. We have changed the title to “Effects of a single joystick-operated ride-on-toy navigation training incorporated into a constraint induced movement therapy program: A pilot feasibility study

Comment 2: I assume you do not have results from children without the ROT program. Because in this program one cannot specify the results of ROT alone. This impacts the introduction and especially the discussion and conclusions.

Response: We completely agree with the reviewer on this point. This was a single group pre-post study design where participants were recruited through convenience sampling (as mentioned in the methods section of the manuscript). Based on the promising results from this pilot study, in the future, we will conduct a larger cross-over trial to assess effects of ROT training. We have revised the introduction and discussion sections to reflect this limitation in study design and have instead acknowledged that the study reflects the combined effects of ROT when incorporated into a CIMT training program. We have also made sure to tone down the conclusions section accordingly.

Comment 3: 11 children joined. They all performed 90 h of CIMT, of which 8 h are ROT. One need to discuss intense the added value based on dose (90 versus 8 h).

Response: We have made sure to clarify the details of training intensity relative to the ongoing CIMT training that children were receiving (see page 5, para 2). Per the reviewer’s suggestion we have acknowledged throughout the paper that the effects are indicative of a combined effect of ROT incorporated into a CIMT training camp. The revised title of the manuscript will also help clarify the focus of the paper upfront for readers.   

Comment 4: Please inform us what is known about the psychometric values of the outcome measures (especially SEM and clinical important change) in the method section and use these values in the results presentation and discussion/ conclusion.

Response: We thank the reviewer for their excellent suggestion. The minimal clinically important difference (MCID) for the arm accelerometry is a change of 575-752 counts (Chen et al., 2018). Please note that MCID values for arm accelerometry are based on data from adult persons with chronic hemiparesis since no similar data are available from children with HCP. At present there are no data available for SEM and MCID values for the SHUEE. However, the SHUEE has been used to assess the efficacy of surgical interventions with children with CP (Lennon et al., 2023; Tedesco et al., 2015; Smitherman et al., 2011; Van-Heest et al., 2015). The percent change values across these studies range from 3.5-9% for the spontaneous functional analysis (SFA) scores and 14.6-44.8% for total dynamic positional analysis (DPA) scores. We have made sure to add this information and discussion regarding measures in the revised version of the manuscript within the methods section (page 3, para 3; page 4, para 2) as well as within the discussion (page 15, para 2).

Comment 4: This is a pilot feasibility and to perform this type of analysis and correlation should be avoided in my opinion. I would like to have the opinion of a statistician.

Response: We thank the reviewer for bringing up this point. We reported correlations between measured outcomes in the original analysis as part of exploratory analyses that sought to understand the nature of relationships between training experiences and children’s performance on standardized tests. Based on the reviewer’s suggestion, we consulted a statistician at our university and they agree that one should be cautious about over-interpreting statistical tests based on a small sample. After understanding our rationale/intention in calculating correlations in this pilot study, they advised us to instead plot the relationships between variables measured in the form of scatter plots that show individual data for readers to visualize the data. They advised us against conducting formal tests of significance for these correlations between variables and instead asked us to report this within the text as exploratory analyses without accompanying tests of significance in view of the small sample size of our pilot study.

In line with these excellent suggestions from the reviewer and our statistical consultant, in the revised manuscript, we have clarified within the introduction that we will merely begin to explore relationships between standardized and training-specific measures within this pilot study (page 2 & 6). We have removed Tables 2A & 2B that previously reported on significance tests for correlations and have instead added scatter plots to highlight associations between all variables based on individual data (see pages 6, 11-13). We have discussed a few promising trends in the data and have acknowledged that given our small sample size, we interpret these data as suggestive of trends for relationships between variables that would need to confirmed using larger samples in our future studies (pages 15-16). We hope the reviewer will find this approach satisfactory.

Comment 5: I prefer the description of the individual scores to indicate improvement or not, without the complexity of these type of statistics, with a low value based on the sample size

Response: We understand the reviewer’s point. Our study was not designed to be a case series. We conducted statistical analyses appropriate for a single group pre-post design. However, given our small sample, we have also reported on effect sizes to indicate the magnitude of improvement (see results section of the manuscript). Per the reviewer’s preference, we have ensured that individual data for each of the measures is represented. Individual data on outcome measures are represented in Figures 2B, 2C, 3B, 3C, 4B and 4C. In addition, as indicated in response to Comment 4, we have also added scatter plots showing individual data on measured variables (pages 12-13).

Comment 6: The paper is written very well. I hope the authors will look into these topics to improve the paper

Response: We thank the reviewer for taking the time to review the manuscript and suggesting valuable edits. We think that the manuscript is significantly improved with the suggested reviewer edits.

Reviewer 2 Report

As a Pediatric Orthopaedic Surgeon, I found interesting your manuscript, despite that it is OUT of my common practice. Dear authors it is important to clarify that the task of joy stick in a ROT uses ONLY flexion of wrist and fingers, permanent pronation and is NOT related with finger extension with the wrist in neutral or extended that is the MAIN problem in hemiplegic hand. The task is not requiring forearm supination, so can be effectively performed. Please comment on this.   CIMT using a cast in the unaffected arm, abolishes the mirror movements, that are IMPORTANT mainly in the severely affected children.

Accelerometers  hardly provide accurate assessment in SMALL range of movements, but your measurements are accurate.

Did you measure any of these children after any type of intervention (botulinum injection or tendon transfer?)  Do you suggest that the joy stick ROT can be part of a CONTINUOUS program, that is essential for all CP children.   Finally hemiplegic children have SEVERE sensory deficit, that is NOT ADDRESSED in your paper. Can you please comment on this.

Author Response

Dear Editor and Reviewers,

We sincerely thank the reviewers for taking the time to review our manuscript and for providing excellent and constructive feedback. We have addressed all concerns raised by both reviewers. We believe that the manuscript is truly strengthened following these revisions. Below, we provided a point-by-point response to reviewer comments with reference to manuscript edits. All edits in the manuscript have been made using track changes.

We hope we have addressed all the concerns of the reviewers and we look forward to the next round of review.

Thank you very much for your time and consideration.

Sincerely,

Authors

Reviewer 2:

Comment 1: As a Pediatric Orthopaedic Surgeon, I found interesting your manuscript, despite that it is OUT of my common practice. Dear authors it is important to clarify that the task of joy stick in a ROT uses ONLY flexion of wrist and fingers, permanent pronation and is NOT related with finger extension with the wrist in neutral or extended that is the MAIN problem in hemiplegic hand. The task is not requiring forearm supination, so can be effectively performed. Please comment on this. 

Response: We understand the reviewer’s point. We acknowledge that we did not address movements of supination, wrist extension, and finger extension in isolation. We focused on functional UE movements and patterns that included wrist extension, finger extension and movements towards supination. We have acknowledged this in the revised manuscript within the methods section. We would also like to clarify that the ride-on-toy navigation program involved 2 components: (1) navigation of the ride-on-toy across different environmental layouts and (2) upper extremity tasks at intermediate stations. We agree with the reviewer that the grasp and operation of the joystick itself typically requires wrist, finger, and hand muscles with the forearm held in pronation. In addition, children require proximal muscles at the elbow and shoulder to control push-pull movements of the joystick in all 4 directions (forward, backwards, right turn, and left turn). In our experience, children with poor UE control tend to also use proximal scapular and trunk muscles to move the joysticks. We incorporated a variety of functional UE tasks into the training program. These UE tasks involved gross motor activities such as reaching in different directions, overhead throwing, pulling, pushing, lifting, and tossing games as well as fine motor activities such as opening and closing, precision grips, picking, sticking, and releasing objects. While singular joint movement and specific isolated movements that are typically limited in HCP were not addressed, the UE tasks promoted multi-joint movements involving shoulder, elbow, forearm, wrist, hand, and fingers including movements of wrist extension, finger extension, and forearm supination. We have added these details of the training program in the revised version of the manuscript (page 5, para 3).

Comment 2: CIMT using a cast in the unaffected arm, abolishes the mirror movements, that are IMPORTANT mainly in the severely affected children.

Response: We understand the reviewer’s point. However, we stayed consistent with principles of CIMT that encourage forced use of the affected UE (Charles et al., 2006; Gordon et al., 2005). We have acknowledged within the limitations section that the CIMT paradigm does not allow for mirror movements (page 16).

Comment 3: Accelerometers hardly provide accurate assessment in SMALL range of movements, but your measurements are accurate.

Response: Accelerometers have been used to document amount and duration of UE activity in both children and adults with hemiplegia (Uswatte et al., 2000; Bailey & Lang, 2012; Lang et al., 2017; Niet et al., 2007; Poitras et al., 2022; Beani et al., 2019; Hayward et al., 2016; Noorkõiv et al., 2014; Dawe et al., 2019; van der paas et al., 2019; as also discussed in the manuscript within the discussion section on page 13, lines 481-485). Moreover, like we discussed in the methods section of the manuscript (page 4, lines 166-168), children wore accelerometers when they were driving the car in a seated position. Therefore, all activity recorded is reflective of their affected UE activity. We acknowledge that accelerometers do not provide information about the quality of movement. We therefore also used video-based measures to code the type/quality of upper extremity activity.   

Comment 4: Did you measure any of these children after any type of intervention (botulinum injection or tendon transfer?)  Do you suggest that the joy stick ROT can be part of a CONTINUOUS program, that is essential for all CP children. 

Response: The children who volunteered for the study had not received a tendon transfer or Botox within the last 3 months (as listed in the inclusion-exclusion criteria within the methods section). We envision the ride-on-toy program to serve as an adjunct therapeutic tool for children with asymmetry in upper extremity function. In our experience, given the engaging nature of the toy itself, children are intrinsically motivated to drive the toy and are more willing to engage in playful UE tasks that are incorporated into the navigation training program. We see the ROT training as an easy-to-use tool that families can integrate into their child’s play routines within real-world environments. Clinicians may also recommend ROTs to families as adjuncts to boost treatment dosing and affected UE use outside conventional therapy settings. However, with regard to the reviewer’s question, we are not able to make claims about the use of ROT for children post-surgery or post-Botox as they were not included within the current study. This would certainly be an area for future study and we hope to pursue such research in our future work. We have acknowledged the lack of generalization of our findings to children with recent histories of surgeries in our limitations section (page 16).  

Comment 5: Finally hemiplegic children have SEVERE sensory deficit, that is NOT ADDRESSED in your paper. Can you please comment on this.

Response: The current manuscript reported on motor outcomes following ROT training incorporated as part of a CIMT program. We agree that we did not report on sensory outcome measures in this manuscript. We acknowledge that children with HCP have sensory deficits that contribute to their clinical presentation. Our intervention focused on providing movement-based navigation training. Children were able to use visual feedback at all times during the training. The reviewer’s point is well taken and we will make sure to include measures of sensory function as outcome measures in our future studies. We have discussed the lack of sensory measures in our test battery as a significant limitation of the study within the revised manuscript (page 16).

Round 2

Reviewer 1 Report

I thank the authors for all changes in the manuscript.

I approve the manuscript in its current form

Author Response

Dear Editor and Reviewers,

We sincerely thank the reviewers and the academic editor for taking the time to review our revised manuscript. We are glad to hear that the reviewers and editor are satisfied with our edits and have only requested for minor modifications of the manuscript. Below, we provided a point-by-point response to editor comments with reference to manuscript edits. All edits in the manuscript have been made using track changes.

We hope we have addressed all the concerns of the reviewers and we look forward to the next round of review.

Thank you very much for your time and consideration.

Sincerely,

Authors

Academic Editor/Reviewer Comments

Reviewer 1:

Comment 1: The addition of the word "effects" in the title that also include "a pilot feasibility study" needs to be improved. We suggest removing the word effects. Perhaps "outcomes" or something like that. We agree there is efficacy for 90 hours of CIMT and this projects add to it and the type of activities that might be included in a camp CIMT treatment.

Response 1: Per the editor/reviewer’s suggestion, we have edited the title of the manuscript. The revised title is, “Outcomes associated with a single joystick-operated ride-on-toy navigation training incorporated into a constraint induced movement therapy program: A pilot feasibility study.”

Comment 2: There seems to be one outlier who had no function at the early time (Figure 3B) that is explained well. Is this the same participant who has their own open dots that are separate from the group in many of the correlation graphs (5A and 5B, one example are the 3 plots under sedentary activities in 5B but there are many)? This single dot is driving the line. This needs to be better explained. We do like the visual analysis and these figures but ask for clearer explanation of the graphs overall and also the interpretation when there is a single dot like this driving some of the plots.

Response 2: We are glad to hear that the reviewers and the editors liked the figures that directly allow readers to visualize raw data from participants. We thank the academic editor/reviewers for noticing the single child who showed a huge improvement in performance. Yes, the same child who has been explained in Figure 3B is the child who has their own open dots separate from the rest of the group in some of the correlation graphs depicted in Figures 5A and 5B. We acknowledge that in some of the graphs, this single child seems to drive the plotted relationship between variables. We would like to clarify that we conducted these analyses that explored preliminary associations between variables (using pooled data and training-related change data) both with and without this child’s data being included as part of these analyses. All the preliminary associations that have been highlighted and discussed further within the narrative of the manuscript hold true irrespective of whether the said child was included or excluded from these analyses. The direction and magnitude of the reported relationships (for both pooled and change data) are similar with and without this child. In fact, within the pooled data, the magnitude of associations were stronger when the child was excluded from the analyses. Moreover, for the graphs in Figure 5B, we only highlight the associations between SHUEE scores and percent time spent in moderate to vigorous activity and no activity bouts; these trends hold true both with and without the child included as part of these analyses. We have made sure to clearly mention within the figure footnotes as well as within the narrative of the manuscript that the trends discussed were consistently seen even if we excluded the child who showed the largest improvement in performance following the training.